# Untargeted Lipidomic Profiling of Dry Blood Spots Using SFC-HRMS

**DOI:** 10.3390/metabo11050305

**Published:** 2021-05-11

**Authors:** Pauline Le Faouder, Julia Soullier, Marie Tremblay-Franco, Anthony Tournadre, Jean-François Martin, Yann Guitton, Caroline Carlé, Sylvie Caspar-Bauguil, Pierre-Damien Denechaud, Justine Bertrand-Michel

**Affiliations:** 1MetaboHUB-MetaToul-Lipidomique, MetaboHUB-ANR-11-INBS-0010, Inserm U1297/Université Paul Sabatier Toulouse III, 31432 Toulouse, France; pauline.le-faouder@inserm.fr (P.L.F.); julia.soullier@inserm.fr (J.S.); anthony.tournadre@gmail.com (A.T.); 2MetaboHUB-MetaToul-Axiom, MetaboHUB-ANR-11-INBS-0010, INRAE Toxalim, Université Paul Sabtier, 31027 Toulouse, France; marie.tremblay-franco@inrae.fr (M.T.-F.); jean-francois.martin@inrae.fr (J.-F.M.); 3MELISA Core Facility, Laboratoire d’Etude des Résidus et Contaminants dans les Aliments (LABERCA), Oniris, INRΑE, 44307 Nantes, France; yann.guitton@oniris-nantes.fr; 4Laboratoire de Biochimie, Hôpital Purpan, CHU Toulouse, 31059 Toulouse, France; caroline.marie.carle@outlook.fr; 5INSERM, UMR1297, Institute of Metabolic and Cardiovascular Diseases, University Paul Sabatier, 31432 Toulouse, France; sylvie.caspar-bauguil@inserm.fr (S.C.-B.); pierre-damien.denechaud@inserm.fr (P.-D.D.)

**Keywords:** Lipidomic 1, Supercritical Fluids Chromatography 3, Plasma 3, Dry Blood Spot 4

## Abstract

Lipids are essential cellular constituents that have many critical roles in physiological functions. They are notably involved in energy storage and cell signaling as second messengers, and they are major constituents of cell membranes, including lipid rafts. As a consequence, they are implicated in a large number of heterogeneous diseases, such as cancer, diabetes, neurological disorders, and inherited metabolic diseases. Due to the high structural diversity and complexity of lipid species, the presence of isomeric and isobaric lipid species, and their occurrence at a large concentration scale, a complete lipidomic profiling of biological matrices remains challenging, especially in clinical contexts. Using supercritical fluid chromatography coupled with high-resolution mass spectrometry, we have developed and validated an untargeted lipidomic approach to the profiling of plasma and blood. Moreover, we have tested the technique using the Dry Blood Spot (DBS) method and found that it allows for the easy collection of blood for analysis. To develop the method, we performed the optimization of the separation and detection of lipid species on pure standards, reference human plasma (SRM1950), whole blood, and DBS. These analyses allowed an in-house lipid data bank to be built. Using the MS-Dial software, we developed an automatic process for the relative quantification of around 500 lipids species belonging to the 6 main classes of lipids (including phospholipids, sphingolipids, free fatty acids, sterols, and fatty acyl-carnitines). Then, we compared the method using the published data for SRM 1950 and a mouse blood sample, along with another sample of the same blood collected using the DBS method. In this study, we provided a method for blood lipidomic profiling that can be used for the easy sampling of dry blood spots.

## 1. Introduction 

Lipids represent a large and complex class of hydrophobic and amphipathic small molecules, with a huge structural diversity (e.g., various combinations of fatty acyls and functional headgroups in phospholipids). They can be divided into six basic groups according to the Lipid MAPS consortium (The LIPID MAPS^®^ Lipidomics Gateway, https://www.lipidmaps.org/ access on 10 May 2021): fatty acyls, glycerolipids, glycerophospholipids, sphingolipids, sterol lipids, prenol lipids, saccharolipids, and polyketides [1]. Changes in the level and/or composition of lipid species and/or classes are highly controlled and occur during physiological processes or after pathological perturbations. Indeed, the dysregulation of lipid metabolism is related to the development of many common diseases, including diabetes, cancers, neurological disorders, and other diseases [2]. Thus, it is crucial to be able to profile the lipidome, i.e., provide a comprehensive and quantitative description of a set of lipid species present in an organism. 

Identifying the global lipid profile in one sample remains a challenge. Due to the structural diversity among lipid molecules, lipidomic profiling is complex [3] and necessitates several approaches and methodologies involving different steps: extraction, separation, and detection. Briefly, the first important step will be to apply a suitable sample preparation for the family of lipids studied: the more hydrophobic family (such as glycerolipids, glycerophospholipids, sphingolipids, and sterols) will generally be extracted through liquid-liquid extraction, while the more hydrophilic species (such as fatty acid metabolites) will be concentrated through a solid phase separation [4]. To profile the least abundant lipids (i.e., free fatty acids, cholesterol, and oxylipids), powerful, targeted, but time-consuming approaches are chosen: liquid chromatography (LC) or gas chromatography (GC) coupled to mass spectrometry (MS) techniques (i.e., LC-MS or GC-MS), which can provide absolute quantitative results under certain conditions [3]. However, to profile the most abundant lipids (i.e., phospholipids, sphingolipids, and triglycerides), high-resolution mass spectrometric approaches are typically used to allow for non-targeted techniques and to propose the largest coverage of the lipidome. These unbiased approaches are usually preferable for large series of samples studied in clinical applications using system biology approaches [5] and can be implemented using different techniques. The lipid extract can be introduced directly in a very high-resolution mass spectrometer detector (usually an Orbitrap) by a shotgun lipidomic, which presents many advantages: it is fast, adapted to low quantities of samples and can be easily automated. Nevertheless, it has quite a few limitations, such as ion suppression, the ambiguous identification of isobaric/isomeric lipid species, and ion source-generated artifacts, which limit its application [6]. The lipid extract can be also separated by liquid chromatography on a polar column (mainly HILIC) to separate the lipids per class [7,8] or on a hydrophobic column, such as C_18_ [9] or C_8_ [10], to separate the molecular species of each lipid class. These chromatographic systems lead to some quite long separations (often between 25 and 45 min), which are doubled due to the need for an injection in positive and an injection in negative mode, because, unfortunately, if some spectrometers allow the alternation of positive- and negative-mode analysis, the heavy data treatment needed in these unbiased methods are not compatible with this mode.

Taking into account all these technical limitations, for untargeted lipidomic profiling, in our facility, we chose to work with a supercritical fluid chromatography system using CO_2_ under pressure as an eluent. This technique is especially suitable for lipids and allows for a shorter separation than liquid chromatography [11,12]. Recently, different studies [8,13] presented the potential of Supercritical Fluid Chromatography (SFC) for profiling lipids in a very short time, especially for clinical data. Once the technique is chosen, the main challenge for an unbiased approach will be the data processing of more than 550 species of lipids. Usually, the first step is the establishment of a database that combines the precise *m*/*z* of each lipid species with the retention time under the selected chromatographic conditions, and then the suitable software has to be optimized to effectively integrate a large quantity of peaks. 

The goal of our work is to develop a new analytical strategy for a high-throughput and comprehensive lipidomic analysis of biological samples applicable for large clinical studies using SFC on a sub-2 μm particle-bridged diethylamine polar column for the separation of a wide range of hydrophobic and hydrophilic lipid classes in one analysis, including the identification and quantitation of individual lipid species using electrospray ionization (ESI)-MS Xevo G2-XS time of flight (QTOF). First of all, our lipidomic method was validated through the analysis of the reference material human plasma, NIST (National Institute of Standards and Technology; SRM 1950), and by comparing it with already published data [14]. Dried blood spots (DBSs) are whole blood collected on filter paper from a simple finger prick, which provides a minimally invasive method for collecting blood samples in nonclinical settings. It represents convenient matrices for collecting and storing human samples, especially for neonates [15,16]. It is less invasive than a classical venous puncture and can be carried out by the patient at home and shipped by regular mail, with no particular risk of contamination [17,18]. The second objective of our work was to evaluate our method using mouse blood samples, comparing whole blood and DBS samples.

## 2. Results and Discussion

### 2.1. Method Development

#### 2.1.1. Optimization of Lipid Class Separation by Supercritical Fluid Chromatography 

In SFC, the most frequently used mobile phase is supercritical carbon dioxide (SCCO_2_), because carbon dioxide can easily be converted into its supercritical state (critical temperature, 31.1 °C; critical pressure, 7.38 MPa) and exhibits favorable properties, such as being nonflammable, chemically inert, relatively nontoxic, easy to handle, and inexpensive. It has a relatively low polarity, which is very similar to n-Heptane, and its polarity can be changed considerably by adding a polar organic solvent, such as methanol, as a modifier. In a previous work, Bamba’s team showed that SFC/MS/MS with a reverse-phase column can be used for the analysis of global lipids, without ESI-incompatible solvents [19]. Holcapek’s team presented a normal-phase silica-based column to perform high-throughput lipidome analysis using SFC [20]. Their lipid class separation using this technique was superior to conventional lipid class separation methods (i.e., NPLC and HILIC) in terms of the analysis time and chromatographic resolution for a wide variety of lipids. The objective here was to obtain the best separation of 17 main classes of lipids: triacylglycerols (TGs), free (FCs) and esterified cholesterols (CEs), diacylglycerols (DGs), free fatty acids (FAs), ceramides (Cers), phosphatidylcholines (PCs), mono hexosylceramides (MHCers), sphingomyelins (SMs), fatty acyl-carnitines (CARs), lysophosphatidylcholines (LPCs), phosphatidylethanolamines (PEs), lactosyl ceramides (LacCers), lysophosphatidylethanolamines (LPEs), phosphatidylglycerols (PGs), phosphatidylinositols (PIs), and phosphatidylserines (PSs). Takeda et al. [13] presented the screening of six different normal-phase columns, including Ethylene-Bridged Hybrid (BEH), 2-Ethyl Pyridine (2-EP), 2-Picolylamine (2-PIC), 1-Aminoanthracene (1-AA), DIOL, and Diethylamine (DEA), which were developed based on 1.7 µm BEH particle technologies, and the DEA column was clearly the most efficient one in separating our lipids of interest. Thus, this column was chosen in our project, and we confirmed this effective separation. Separation parameters, such as the modifier, gradient, and temperature of the column, were then optimized on our instrument to obtain the most intense signal (data not shown). First of all, the modifier was optimized: a mixture of methanol, 1% of water (with 20 mM ammonium acetate), and 0.1% of ammonia was used as eluent B. Despite the great chromatographic separation of the lipids family and an excellent detection of PSs (which is usually very weak using SFC), the use of ammonia was abandoned, because it generated a significant overpressure after several injections, which led to the method having a non-robust reliability. We optimized the quantity of water to improve the elution of hydrophilic lipids (PIs, PSs): 0.5, 1, 2, 3, and 4% of water in methanol have been tested, and the best result was obtained with 2%. 

In these conditions, we estimated that the separation of our 17 classes of lipids was optimal, with a return to the base line from the two peaks. Figure 1 presents the Total Ionic Current (TIC) (Figure 1A) of the mixture of the 17 standards and the extracted ion chromatogram (EIC) (Figure 1B) for each internal standard, with the addition of free cholesterol (FC) and lauroyl carnitine (the internal standard (ISTD) for these classes were not exploitable).

#### 2.1.2. Optimization of Lipid Class Detection

The identification of molecular ions was performed on the 15 internal standards, free cholesterol, and lauroyl carnitine, corresponding to the different lipid classes described below in positive and negative ionization mode (Appendix A). The presence of multiple adducts, such as [M+H]^+^, [M+NH_4_]^+^, in positive ionization mode and [M-H]^−^ and [M+CH_3_COO]^−^ in negative ionization mode were mainly observed for the different lipid classes. Nevertheless, we observed that sodium adduct [M+Na]^+^ and in-source fragments [(M+H)-H_2_O]^+^ constituted the majority molecular ions for CEs and ceramides, respectively. The addition of ammonium acetate in the modifier solvent improved the signal of TGs and cholesteryl esters, but it did not allow the CEs sodium adduct to be decreased. In parallel, a loss of signal intensity was observed when ammonium acetate was added in the make-up solvent, without inverting the intensities of the adducts, [M+NH_4_]^+^ and [M+Na]^+^, for CEs. Moreover, the detection of CEs was quite difficult because of the low ionization efficiency. The addition of 5% of water to the make-up solvent drastically increased the signal. Finally, the MS cone voltage was optimized to decrease the in-source fragmentation obtained for numerous lipid classes.

The results obtained in our optimized conditions are presented in Table 1. In positive ionization mode: esterified cholesterols were detected as [M+Na]^+^ adducts, TGs as [M+NH_4_]^+^, PCs, LPCs, SMs, PEs, LPEs, fatty acyl-carnitines, and lactosyl ceramides as [M+H]^+^, and finally, DGs, Cers, MonoHexosyl ceramides, and free cholesterols were detected as [(M+H)-H_2_O]^+^. In negative ionization mode: free fatty acids, PGs, PIs, and PSs were all detected as [M-H]^−^ ions.

Table 1 presents the retention times obtained for ISTD and a few species for each class.

#### 2.1.3. In-House Database Development

The total characterization of the different internal standards belonging to each lipid class were performed using MS/MS experiments. We validated each structure in terms of the presence of specific daughter ions (Table 1). For instance, phosphatidylcholine and sphingomyelin were characterized using the PC ion at *m*/*z* 184.0726 [21] and ceramides and hexosyl ceramide using the sphingoid base backbone at *m*/*z* 264.2682 [22]. Moreover, we validate the characterization and retention time of each lipid class through the analysis of different lipid subspecies (Table 1). The huge advantage of working with a DEA column is the rapid and efficient chromatographic separation by lipid class due to the SFC properties. Thus, thanks to the high mass accuracy of the QTOF mass spectrometer and the fully chromatographic separation of the lipid class, the identification of other subspecies of each family becomes easy and unambiguous. Based on the molecular adducts, the Lipid Maps database, and human plasma lipidic extract characterization, we developed an in-house database containing a unique mass to charge ratio and a retention time for lipid subspecies. This was achieved through a manual search for each compound in a plasma lipidomic extract, with a tolerance of 10 ppm, and the verification of the isotopic profile and adducts. The metabolomic guidelines [23] were used to determine the level of identification of different lipids. Thus, using commercially pure internal standards (and free cholesterol), MS and MSMS experiments were performed, allowing us to annotate them at level 1. Metabolites that are characterized by their *m*/*z* ratio and their retention time according to their lipid class and that do not have insaturation in the fatty acyl chain are annotated at level 2. Finally, metabolites characterized by their *m*/*z* ratio and their retention time according to their lipid class and that have insaturation in the fatty acyl chain are annotated at level 3. For example, in our chromatographic method, isomeric species, such as PC 18:0/16:1 and PC 18:1/16:0, are not separated (data not shown). Thus, in our database, we annotate these lipids as a single species, including the total number of carbon and double bonds, such as PC 34:1, and assign them to level 3 for identification. Finally, our in-house database contains 503 lipids belonging to the 17 main class of lipids, which are as follows: 28 cholesteryl esters and free cholesterols, 60 TGs, 63 DGs, 34 Free Fatty Acids, 36 ceramides, 84 PCs, 7 mono hexosyl ceramides (galactosyl and glucosyl ceramides are not separated in our chromatographic conditions), 27 SMs, 25 LPCs, 11 fatty acyl-carnitines, 72 PEs, 3 LacCers, 8 LPEs, 21 PGs, 16 PIs, and 7 PSs (Appendix A).

#### 2.1.4. Linearity and Sensitivity 

Due to the lack of pure standards and the differences in the MS detection of the studied species, the quantification of the complex lipids studied remains complicated. To obtain an idea of the linearity and sensitivity of the QTOF detector, we performed some calibration lines only on ISTD with different concentrations, ranging from 244 pg/µL to 250,000 pg/µL, in triplicate. The results are summarized in Table 2. The repeatability for the retention time is very good (between 0 and 0.2%) for all classes, except for TGs and CEs. The peaks for these two classes are larger (0.2 min) than the other one (0.08 min), so with a high concentration, the peak apex shifts slightly, which allows for a large RSD. The range of linearity is quite low, which is typical of QTOF detectors, and different between classes. The calibration range quickly reaches a maximum for CEs, DGs, TGs, and Cers (under 10 µg injected), and it is better for LPCs, PCs, SMs, MHCs, CARs, and FAs (between 10 and 100 µg injected) and much better for PEs, LPEs, PGs, PIs, and PSs (between 100 and 250 µg injected). Most of the standards were detected at the lowest concentration, with coefficients of determination (r^2^) ranging from 0.979 to 0.999. We should note that like an ESI source, free cholesterol is not well detected at a low concentration: a minimum of 3.9 ng needs to be injected to allow for a detection; however, above this value, the detection is linear. The repeatability was correct (RSD < 10%), except for FAs, Cers, and CEs (between 20% and 40%).

### 2.2. MS-DIAL Processing Workflow

Data processing was performed using the MS-DIAL software [24]. Raw Waters data were first converted to analysis base files (abf) in order to create a MS-DIAL project. As the Waters QTOF data have a supplementary function, corresponding to the lock spray analysis, a package must be downloaded on the Waters web site and installed in the abf converter software. To process our data in MS-DIAL, two projects are necessary: one in positive ionization mode for the identification of CEs, TGs, Cers, MHCers, LacCers, SMs, PCs, PEs, LPCs, LPEs, and CARs and one in negative ionization mode for the identification of FAs, PGs, PIs, and PSs. Numerous parameters can be changed in MS-DIAL for data collection, peak picking, and identification, as well as alignment. Based on the mass accuracy of the QTOF mass spectrometer and the repeatability of the retention time between the analyses, all parameters were carefully optimized. All MS-DIAL parameters are described in Appendix A. Nevertheless, we observed numerous errors in the integration of the peaks belonging to the CE and TG classes. The key parameter in our data processing was the mass slice width, displayed in the peak collection window. The errors made in integration could be due to the important abundance of CEs and TGs in plasma, but it may also be due to the peak tailing observed in the chromatogram (Figure 1). Thus, in order to correct the problem, two methods of analysis parameter settings have been developed for the positive ionization mode. We used our in-house database for the identification of lipids, after a careful optimization of the MS tolerance to precisely determine each retention time. 

### 2.3. Analysis of NIST Reference Plasma

To evaluate the performance of our new SFC method, we decided to analyze the National Institute of Standards and Technology Standard Reference Material 1950—Metabolites in Frozen Human Plasma (SRM 1950). This plasma was constructed from 100 fasted individuals within the age range of 40 to 50 years, which represented the average composition of the US population (COA, www.nist.gov/srm, (accessed on 5 January 2015)). The analysis, conducted by a consortium of 31 international laboratories, allowed consensus values to be obtained and qualitative profiling of the main lipid classes to be conducted [14]. It is recommended that this certified reference material is used to aid in standardization and quality assessment to validate a new method, at least until new reference materials are created. The extraction method chosen was a modified Folch liquid-liquid extraction [10], where the aqueous phase and the organic phase were pooled together after centrifugation, allowing the best recovery for more hydrophilic lipids, such as PSs, PIs, or gangliosides, to be obtained. This was performed on 10 µL of plasma in the presence of an internal standard mixture. Our first idea was to use the SPLASH^®^ LIPIDOMIX^®^ Mass Spec Standard, specially developed for plasma analysis, which includes all of the major lipid classes at ratios relative to human plasma, but we noticed that a few molecules were not stable after a few injections, so we decided to prepare our own mixture of 15 internal standards, with one from each of the lipid classes analyzed, except for FCs and CARs. For FCs, the quality of cholesterol d7 (deuterated on lateral chain) was not pure enough to be used, so ISTD CE 17:0 was related to FCs. CARs were not foreseen in the method at the beginning, but they were added afterwards, so their ISTD was not present in our extracts, and they were quantified with the ISTD, with the closest retention time being that of LPC 11:0. The absolute quantification of complex lipids using mass spectrometry detection is complicated [25]. For this reason, we decided to compare our data to the published ones from a qualitative point of view for the 10 detected lipid classes: Sterols, TGs, FAs, Cers, SMs, PCs, PEs, PIs, LPEs, and LPCs. For these 10 classes, we detected more species (Appendix A) than in the reference publication [14], so we focused the comparison only on the common species. For each class of lipid, the percentage of each species within the class were calculated and compared on a radar chart (Figure 2).

Overall, the results were very comparable. However, DGs, PSs, PGs, MHCers, and LacCers were very slightly detected in our extract, yet the ISTD for these classes were very well detected: apparently in our conditions, with only 10 µL of plasma NIST extracted, we were not sensitive enough to catch these classes. Concerning CARs, we were able to measure 7 different species, but they were not measured in the reference paper, so we did not add them to the comparison.

### 2.4. Assessment of the New SFC Method Using the Dry Blood Spot (DBS) Method

Dry Blood Spots have gained a broad use in medicinal chemistry and toxicokinetic research, as well as in lipidomics [26] and metabolomics studies. It is a sampling method that is especially practical in clinics. For these reasons, we decided to test our new method using this matrix and by comparing it with blood [27,28]. We used mouse blood to test our analysis. The extraction protocol was very similar to the one used for plasma: the blotting paper punches (which correspond to approximately 2.5 µL of deposited blood) were simply mixed in the Folch modified mixture [10]. It is important to note that the ISTD mixture was deposited in the tube and not on the paper. The extractions were conducted on one or two punches. One punch is enough to get a complete profile (data not shown), so all our studies were conducted only on one punch. 

#### 2.4.1. Lipidomic Profiling of Whole Blood vs. Dry Blood Spots

With DBS (called DBS_T0), 2.5 µL of whole blood was compared. The samples were extracted immediately after the blood deposit in a tube or on the blotting paper. It is important to note that the reproducibility of whole blood extract is not convenient, because the blood coagulates quickly, as soon as it is deposited in a tube. It is then more complicated to obtain an efficient homogenization during the sample preparation. The relative quantities of 13 detected lipid classes were compared using a box plot (Figure 3, panel A to M). 

Globally, we noticed that the relative quantities were different between the whole blood extract (dark blue) and DBS (medium blue), especially for phospholipids (PCs, PIs, PEs, and LPCs), TGs, and CEs. Two factors can explain these differences: first, the quick coagulation of blood, which can strongly modify the yield of extraction, and second, the ISTD can be deposited directly in the tube for DBS (and not on the blotting paper), which allows for a better extraction, leading to a decrease in the relative quantification (inversely proportional to the area of the ISTD). Importantly, DBS extraction is much more reproducible than the whole blood one for PLs (especially for PIs, PCs, PEs, and LPEs), TGs, and CEs. For Cers, SMs, and CARs, the extractions remain very similar. However, it is important to note that the FAs present an unusually high level of detection, which is not typical. In fact, if the level of FAs in the blank sample (a punch of paper without any deposit) is very high, then the relative quantification of FAs in the DBS is not usable in these conditions. For all the other lipid classes, despite the differences in the relative quantification, the distribution of the different lipid classes in relation to the total is very similar: the profiles compared in Figure 3 (panel N) are very close. 

The qualitative profiles within each class were checked by controlling the detection of 168 different species (Appendix A). The molecular species profiling was very similar for almost all the classes. The results are presented for Cers (Figure 4 Panel A) and LPCs (Figure 4 panel B) in Figure 4, even for minor species. 

Considering the length of the acyl chain for TGs, CEs, PEs, PCs, and FAs, the detected profiles were different, depending on the method used (Figure 4 panel C). For instance, for PCs, shorter chain species (32 carbons) were extracted more efficiently in the DBS than in the whole blood, and the opposite is the case for longer chains (more than 40 carbons). Nevertheless, the general qualitative profile remains the same (Figure 4).

#### 2.4.2. Storage Effect on Lipidomic Profiles

One of the advantages of DBS samples is the easy transportation of the sample, but in this condition, it is important to assess the lipid stability over a wide range of temperatures during storage and transportation. After discussion with clinicians, we decided to choose the worst situation with the storage of DBS 3 weeks at room temperature indeed these sample are not usually stored in fridge. So finally, we compared the results obtained for DBS extracted on the day of the deposit (DBS_T0; Figure 3 in medium blue) versus DBS extracted 3 weeks after storage at room temperature in dark (DBS_T3; Figure 3 in light blue). The distribution of the main lipid classes within the total were similar (Figure 3, panel N), which was confirmed regarding the relative quantification of PCs, PEs, PIs, SMs, LPEs, and LPCs on a boxplot (Figure 3). Unfortunately, we were not able to explain the increase in TGs and ceramides after 3 weeks of conservation, although this class of lipids should not be accumulated in a matter of weeks.

The qualitative aspects of the molecular species within each class were compared: they were very close for all classes (Appendix A), as shown in the case of Cers or LPCs (Figure 4 panel A and B). A few classes, such as PC (Figure 4 Panel C) and PE shorter species (between 32 and 34 carbon atoms), are well detected after 3 weeks, while longer species (between 36 and 40 carbon atoms) tend to be less well detected.

## 3. Materials and Methods

### 3.1. Chemical and Reagents

Methanol, Chloroform, Isopropanol, and Plasma NIST were purchased from Sigma Aldrich (Saint Quentin Fallavier, France). Acetonitrile (ACN; HRMS grade) and formic acid (HRMS grade) were purchased from Thermo Scientific. The water used in this study was purified using a milliQ system (Millipore). The internal standards (ISTD) (see Appendix A) used in this development were acquired from Avanti Polar Lipids, Inc. (Alabaster, AL, USA), except FA 17:0, which was purchased from Sigma Aldrich (Saint Quentin Fallavier, France). The LIPID MAPS nomenclature was followed throughout this manuscript [29]. 

### 3.2. Animals/DBS Prelevement 

The mice used in this study were C57B6 control mice. All the experimental procedures were approved by our institutional animal care and use committee, CEEA122 (#7796-2016112509262218), and conducted according to the Inserm guidelines and the 2010/63/UE European Directive for the care and use of laboratory animals. Blood was collected at the tail vein. For typical blood samples, blood was collected using EDTA-coated minivette (17.2113.050, Sarstedt) and transferred to a 0.5 mL tube for blood-lipidomic analysis. For the Dry Blood Spot (DBS) samples, a drop of blood was directly applied to Whatman 903^TM^ paper (#10531018, GE Healthcare). The blood strain needed to be visible on the back of the whatman paper. A punch corresponding to 2.5 µL of blood was used for lipidomic analysis, which was conducted on the day of the blood collection (DBS-T0) and repeated 3 weeks after room temperature conservation (DBS-T3). All the samples were prepared in triplicate (*n* = 3). 

### 3.3. Sample Preparation

Lipids were extracted according to a modified Folch extraction [10,30]. Briefly, samples (10 µL of plasma NIST or 2.5 µL of blood and an equivalent 2.5 µL of blood punched on the DBS), water (100 µL), CHCl_3_:MeOH (490 µL; 50:50; *v*/*v*), and 10 µL of the Internal Standard (ISTD) mixture (Appendix A) were added to an Eppendorf. They were agitated for 1h at room temperature, and 75 µL of water was added in the Eppendorf. They were agitated for 1 min and centrifuged at 4000 rpm for 10 min at 4 °C. The upper phase was recovered and dried under a nitrogen flow. The lower phase was pooled to the dried aqueous phase and concentrated under nitrogen. The samples were reconstituted with 100 µL of MeOH:IPA:H_2_O (65:35:5; *v*/*v*/*v*). The Plasma NIST (SRM 1950) extraction was performed 10 times (*n* = 10). The DBS and blood were extracted 5 times each (*n* = 5).

### 3.4. Supercritical Fluid Chromatography Conditions

The Ultra-Performance Convergence Chromatography (UPC^2^) was coupled on-line to an Xevo G2-XS time of flight (Qtof) (Waters, Milford, MA, USA), equipped with ESI. The analysis was performed in both ionization modes (positive and negative) in two separate runs for major lipid classes detection at a resolution of 16,000 (at *m*/*z* 400). Then, 1 µL of lipid extract was injected into the ACQUITY UPC^2^ Torus diethylamine (DEA) column (100 × 3.0 mm inner diameter (i.d.), particle size: sub-1.7 µm, Waters) at 40 °C. Mobiles phases with a flow rate of 1.2 mL/min were constituted by SCCO_2_ for the A phase and MeOH:H_2_O (98:2; *v*/*v*), with 20 mM of ammonium acetate for the B phase (modifier). The gradient program was as follows: the initial conditions were 1% of B solvent; from 0.5 min to 6 min, this was increased to 40%; then from 6 min to 6.10 min, it was increased to 65%. B was maintained at 65% for three minutes, then the gradient was brought back to initial conditions for three minutes using an active back pressure regulator (ABPR), with 1.500 pounds per square inch (psi). From 6 to 9 min, the flow rate was decreased to 1.1 mL/min. The make-up was MeOH:H_2_O (95:5; *v*/*v*) at 0.1 mL/min for all runs.

### 3.5. Mass Spectrometry Parameters

The source parameters were set as follows: for the positive and negative analysis source, the temperature was 150 °C; the capillary voltage was at −2.6 kV in negative mode and 3 kV in positive mode; the desolvation gas flow rate was 1000 L/Hr; the cone gas flow was set to 50 L/Hr; and the desolvation temperature was 550 °C. The analyses were performed in MS full scan in centroid mode from 50 to 1500 Da, with dynamic range extended (DRE) activated.

MS/MS experiments were performed in positive and negative ion modes using the same instrument. The targeted MSMS method was developed using a ramp of collision energy from 10 to 50 eV. The isolation width was set to *m*/*z* 5. The MSMS mass spectra were inspected manually in order to confirm the annotations. All identifications were conducted with a mass precision of 10 ppm for the MS and MS/MS experiment. 

### 3.6. Measurement of Linearity and Sensitivity

The calibration curves were prepared using the ISTD (List on Appendix A) solubilized in 75 µL of MeOH:IPA:H_2_O (65:35:5; *v*/*v*/*v*). The ISTD were diluted to obtain 11 different concentrations, ranging from 244 pg/µL to 250 000 pg/µL (final concentration), and 1 µL was injected (*n* = 3). The repeatability of the areas and retention times was obtained using the relative standard deviation (RSD, %) of the peak areas and retention time: RSD = (standard deviation/mean) * 100. The ranges were processed using the TargetLynx application in the MassLynx software (waters).

### 3.7. Data Analysis/Normalization 

The UPC^2^-QTOF was controlled using the MassLynx software, version 4.1. The data files (.raw) were converted into .abf using an analysis base file converter. The quantification and identification were performed using MS-DIAL, version 4.85, with the import of the .abf data files. The data processing with MSDIAL was realized in an in-house database, including the molecule, mass to charge ratio, retention time, and selected adduct. Three methods were created to process the lipids. The first one was used for FAs, PGs, PIs, and PSs, the second for Ceramides, PCs, mono hexosyl ceramides, SMs, fatty acyl-carnitines, LPCs, PEs, lactosyl ceramides, and LPEs, and the last one for CEs, TGs, and DGs. All peak areas were extracted from MS-DIAL [28]. The areas were normalized by the ISTD areas of each family and multiplied by the quantity of ISTD injected (see Appendix A) to obtain the relative quantity. The percentages of each lipid class were calculated with the sum of the relative quantities of each class divided by the sum of all families. The percentages of each molecular species within each class were calculated using the relative quantity of one lipid divided by the sum of the relative quantity from all the considered families. A boxplot was used for graphically depicting the relative quantification of the main lipid classes through their quartiles. The rectangle spans from the first quartile (lower edge) to the third quartile (upper edge). The segment inside shows the median (with the box divided into 2 equal parts). Lines extending vertically from the box (whiskers) indicate the minimum and maximum values (excluding outliers). The outliers are plotted as individual points (red stars). The boxplot illustrates the spread of continuous features.

## 4. Conclusions

Lipids are key metabolites in several physiological mechanisms, so large and rapid profiling is very important in obtaining them, especially in clinical applications. Due to the diversity and complexity of these molecules, unbiased lipidomic profiling is a real challenge. In this publication, we aimed to develop a rapid and robust technique using Supercritical Fluid Chromatography, coupled with the high-resolution mass spectrometry detector, QTOF. First, 17 different lipids class were separated on a polar DEA column, the conditions of source ionization and detection were optimized to obtain the best sensitivity, and an in-house database was built to interrogate the mass spectrometry profiles using MS-DIAL. The method was validated on the reference material, NIST plasma (SRM 1950), with a comparison with the published data. The 10 class qualitative profiles obtained using the new methods were very close to the references values. For 4 classes (DGs, PSs, mono hexosyl ceramides, and lactosyl ceramides), our method was not sensitive enough to detect them. 

Finally, we used the DBS method, which is a convenient matrix for collecting and storing human blood samples. The lipidomic profiles were compared using whole mouse blood and DBS extracted on the day of the blood collection (DBS_T0) and 3 weeks after the blood collection (DBS_T3). Overall, the relative quantification of each class was slightly different, but the proportion of different classes within the total lipid content remained quite close, even after 3 weeks, except for free fatty acids, which were very important in blank extraction. The qualitative profile in each class is pretty good, except that for TGs and CEs, but the general profile is still very close. While the spotting and drying of the blood have slightly distinct effects on the lipid profile of a certain class, it is still possible to obtain information on global lipid metabolism changes, which is comparable to the lipid profiles obtained from plasma collected through venipuncture. We really do hope this method will be used in the clinical profiling of lipids for large studies on plasma samples to provide lipidomic results that usually require 4 to 6 different methods and on DBS, especially for phospholipids, sphingolipids, and fatty acyl-carnitines, even after 3 weeks of storage.

## Figures and Tables

**Figure 1 metabolites-11-00305-f001:**
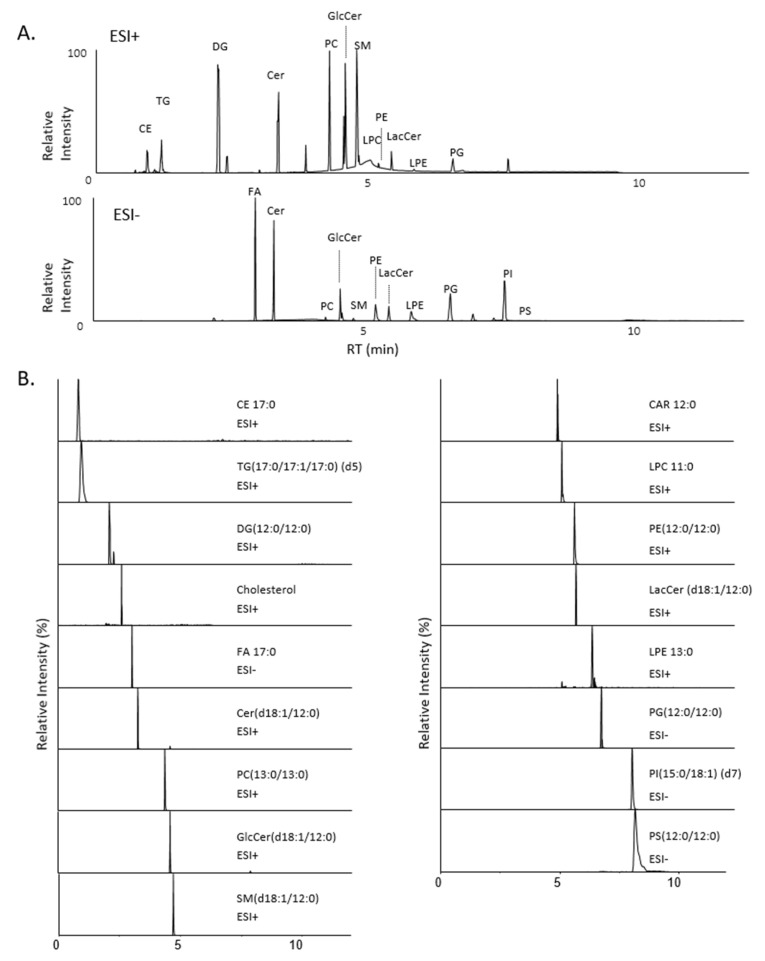
(**A**) Total ion chromatogram (TIC) of internal standards mixture in positive (up) and negative (down) ionization mode. (**B**) Extracted ion chromatogram (EIC) for each internal standards plus free cholesterol and AcylCarnitine: CE 17:0; TG(17:0/17:1/17:0) (d5); DG(12:0/12:0); Cholesterol; Cer(d18:1/12:0); FA 17:0; GlcCer(d18:1/12:0); SM(d18:1/12:0); CAR 12:0; LPC 11:0; PE(12:0/12:0); LacCer(d18:1/12:0); LPE 13:0; PG(12:0/12:0); PI(15:0/18:1)(d7); PS(12:0/12:0) detected either in the positive (ESI+) or in the negative (ESI-) ionization mo.

**Figure 2 metabolites-11-00305-f002:**
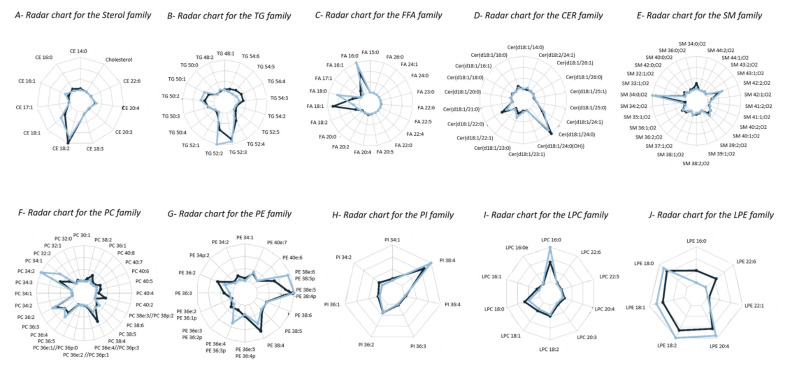
Radar chart of % of molecular species per lipid class generated for SRM195 published data (medium blue line) compared to % molecular species per class of lipid generated for data obtained with the new SFC method (dark blue line). (**A**) Sterol, (**B**) Triacylglyceride (TG), (**C**) Free fatty acid (FFA), (**D**) Ceramides (Cer), (**E**) Sphingomyelin (SM), (**F**) Phosphatidylcholine (PC), (**G**) Phosphatidylethanolamine (PE), (**H**) Phosphatidylinositol (PI), (**I**) Lyso-phosphatidylcholine (LPC); (**J**) Lyso-phosphatidylethanolamine (LPE).

**Figure 3 metabolites-11-00305-f003:**
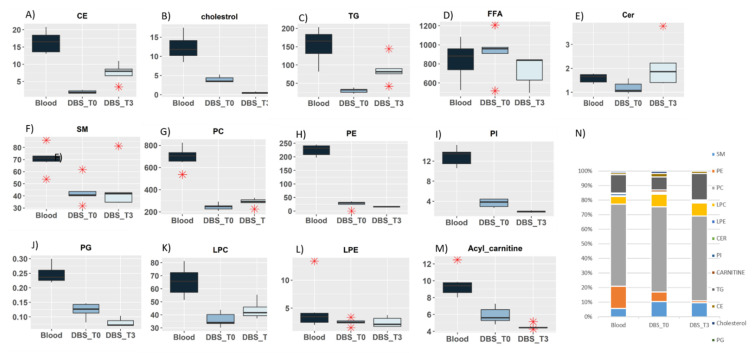
(**A**–**M**) Box-cum-whisker plots showing relative quantification for main lipid classes detected in Blood (dark blue), the Dry Blood Spot (DBS) T0 extracted the day of the deposit on blotting paper (medium blue) and DBS T3: 3 weeks after the day of deposit (light blue). In the box plots, the boxes denote interquartile ranges, horizontal line inside the box denote the median, and bottom and top boundaries of boxes are 25th and 75th percentiles, respectively. Lower and upper whiskers are 5th and 95th percentiles, respectively. (**N**) Histogram of % of lipid classes within the total lipids computed for Blood, Dry Blood Spot (DBS) T0 and Dry Blood Spot (DBS) T3 samples.

**Figure 4 metabolites-11-00305-f004:**
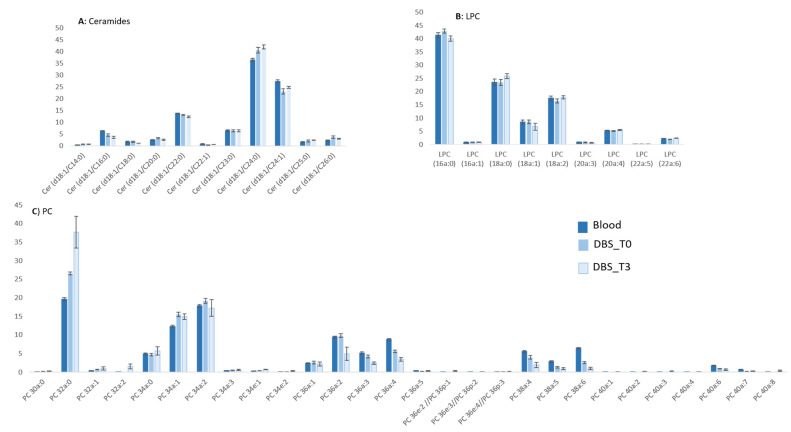
Relative quantities (in %) of each molecular species within each class of whole blood extract (dark blue), Dry Blood Spot (DBS) T0 extracted the day of the collection (medium blue) and Dry Blood Spot (DBS)T3 extracted 3 weeks after collection (light blue) of (**A**) Ceramide; (**B**) Lysophosphtidylcholine (LPC); (**C**) Phosphatidylcholine (PC).

**Table 1 metabolites-11-00305-t001:** Chromatographic and mass spectrometry characterisation with theorical (Th) and experimental (Exp) masses with the *m*/*z* of daughter (daug) ion and their retention time (RT) for internal standard and few molecular species.

Lipid Class	Species	RT (min)	Adduct	*m*/*z* Exp	*m*/*z* Th	Mass Dev (ppm)	*m*/*z* Daug Ion Exp	*m*/*z* Daug Ion Th	Mass Dev (ppm)	Structure of Daug Ion
Choesteryl ester	CE 17:0	0.94	[M+Na]^+^	661.5886	661.5894	1.2	no frag			
Free cholesterol	Chol	2.4	[(M+H)-H_2_O]^+^	369.3522	369.3516	−1.6	147.1152	147.1138	−9.5	C_11_H_15_^+^ (ring A and B)
Triglycerides	TG(17:0-17:1-17:0) (d5)	1.05	[M+NH_4_]^+^	869.8320	869.8329	1.0	582.5522	582.5496	−4.5	[(M+H)-FA]+
	TG 46:0	1.04	[M+NH_4_]^+^	796.7406	796.7389	−2.2	523.4724	523.4721	−0.6	[(M+H)-FA]+
Diglycerides	DG(12:0/12:0)	2.19	[(M+H)-H_2_O]^+^	439.3789	439.3782	−1.6	183.1751	183.1748	−1.6	[(FA+H)-H_2_O]^+^
	DG 32:0	2.33	[(M+H)-H_2_O]^+^	551.5043	551.5034	−1.6	239.2376	239.2375	−0.4	[(FA+H)-H_2_O]^+^
Free fatty acid	FA 17:0	3.01	[M-H]^−^	269.2494	269.2486	−3.0	no frag			
Ceramides	Cer(d18:1/12:0)	3.19	[(M+H)-H_2_O]+	464.4469	464.4462	−1.5	264.2642	264.2682	15.1	[(Sphingosine+H)-2H_2_O]^+^
	Cer(d18:1/24:0)	3.27	[(M+H)-H_2_O]^+^	632.6313	632.6340	4.3	264.2642	264.2682	15.1	[(Sphingosine+H)-2H_2_O]^+^
Phosphatidylcholine	PC(13:0/13:0)	4.36	[M+H]^+^	650.4772	650.4755	−2.6	184.072	184.0726	3.3	phosphocholine ion
	PC(16:0/18:1)	4.34	[M+H]^+^	760.5847	760.5851	0.5	184.072	184.0726	3.3	phosphocholine ion
Mono Hexosyl Ceramide	GlcCer(d18:1/12:0)	4.58	[(M+H)-H_2_O]^+^	626.4973	626.4990	2.7	264.2673	264.2682	3.4	[(Sphingosine+H)-2H_2_O]^+^
	GalCer(d18:1/16:0)	4.52	[M+H]^+^	700.5751	700.5722	−4.1	264.2707	264.2682	−9.5	[(Sphingosine+H)-2H_2_O]^+^
Sphingomyelin	SM(d18:1/12:0)	4.68	[M+H]^+^	647.5108	647.5123	2.2	184.072	184.0726	3.3	phosphocholine ion
	SM(d18:1/18:0)	4.66	[M+H]^+^	731.6033	731.6062	3.9	184.072	184.0726	3.3	phosphocholine ion
Fatty AcylCarnitine	CAR 12:0	4.78	[M+H]^+^	344.2791	344.2795	1.2	183.1752	183.1748	−2.2	[(FA+H)-H_2_O]^+^
Lysophosphatidylcholine	LPC 11:0	4.93	[M+H]^+^	426.2642	426.2615	−6.3	184.072	184.0726	3.3	phosphocholine ion
	LPC 20:0	4.83	[M+H]^+^	552.4035	552.4024	−2.1	184.072	184.0726	3.3	phosphocholine ion
Phosphatidylethanolamine	PE(12:0/12:0)	5.21	[M+H]^+^	580.3982	580.3973	−1.6	439.3797	439.3782	−3.4	[(M+H)-phosphoethanolamine -H_2_O]^+^
	PE(16:0/16:0)	5.29	[M+H]^+^	692.5219	692.5225	0.9	551.5028	551.5034	1.1	[(M+H)-phosphoethanolamine -H_2_O]^+^
Di Hexosyl Ceramide	LacCer(d18:1/12:0)	5.54	[M+H]^+^	806.5635	806.5624	−1.3	264.2707	264.2682	−9.5	[(Sphingosine+H)-2H_2_O]^+^
Lysophosphatidylethanolamine	LPE 13:0	6.21	[M+H]^+^	412.2485	412.2459	−6.4	271.2288	271.2268	−7.4	[(M-H)-ethanolamine]^−^
Phosphatidylglycerol	PG(12:0/12:0)	6.6	[M-H]^−^	609.3776	609.3773	−0.5	199.1721	199.1704	−8.5	RCOO^−^
Phosphatidylinositol	PI(15:0/18:1) (d7)	7.84	[M-H]^−^	828.5634	828.5625	−1.1	288.2911	288.2919	2.8	RCOO^−^
	PI(18:1/18:1)	7.89	[M-H]^−^	861.5508	861.5499	−1.0	281.2503	281.2486	−6.0	RCOO^−^
Phosphatidylserine	PS(12:0/12:0)	8.25	[M+H]^+^	622.3727	622.3726	−0.2	535.3327	535.3405	14.6	[(M-H)-serine]^−^
	PS(18:0/18:0)	7.92	[M-H]^−^	790.5517	790.5604	11.0	703.5201	703.5283	11.7	[(M-H)-serine]^−^

**Table 2 metabolites-11-00305-t002:** Evaluation of the repeatability, retention time variation, and linearity (from 250 to 250,000 pg µL^−1^) of the 15 internal standards, representing the six lipid categories.

Metabolites	Repeatability (RSD, %)	Retention Time Variation (RSD, %)	Linearity r²	linearity (pg/µL)
Cholesterol	2.6	0.4	0.997	3906.25–250,000
CE 17:0	41.1	1.3	0.94	250–1953
DG(12:0/12:0)	4.2	0.7	0.98	250–3906
TG(17:0/17:1/17:0d5)	6.1	1.3	0.99	250–7813
Cer(d18:1/12:0)	13.4	0.2	0.99	250–7813
LPC 11:0	11.0	0.2	0.97	250–31,250
PC(13:0/13:0)	5.4	0.0	0.99	250–31,250
SM(d18:1/12:0)	8.2	0.2	0.99	250–31,250
GlcCer(d18:1/12:0)	6.7	0.1	0.99	250–31,250
CAR 12:0	27.0	0.0	0.966	250–31,250
FA 17:0	23.6	0.0	0.99	250–31,250
LacCer(d18:1/12:0)	2.2	0.3	0.99	250–62,500
PE(12:0/12:0)	6.0	0.2	0.99	250–125,000
LPE 13:0	0.8	0.2	0.99	250–125,000
PG(12:0/12:0)	10.0	0.1	0.99	250–125,000
PI(15:0/18:1d7)	2.1	0.1	0.99	250–125,000
PS(12:0/12:0)	4.6	0.2	0.99	25–125,000

## Data Availability

Raw data were deposit on EBI Metabolight web site (www.ebi.ac.uk/metabolights/, accessed on 10 May 2021) with the study number MTBLS2820.

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
