# Peer review of "Untargeted Lipidomic Profiling of Dry Blood Spots Using SFC-HRMS"

_metabolites, 2021, doi:10.3390/metabo11050305_

Round 1

Reviewer 1 Report

Le Faouder et al report a novel application of Supercritical Fluids Chromatography coupled with MS for global untargeted detection of lipids in blood samples.

I think the method is well described and supported by adequate data demonstrating that it is suitable for blood lipidomic profiling.

Minor points.

1) It seems to me that the manuscript is a little vague in the precise criteria for metabolite identification. Supplemental table 2 mentions an identification score. How is this calculated ?

What is the mass accuracy in MS and MSMS mode for identifying metabolites ?

MSMS spectra of internal standards are missing.

Could the authors provide a list of specific daughter ions for the internal standards that were utilised in the identification.

2) Line 262 -268. Would it be possible to calculate estimates in sensitivity and specificity for detecting lipid classes by the authors method ?

The authors note that some lipids were more difficult to detect with the novel method. Is that due to differences in lipid extraction methods or due to the novel Supercritical Fluids Chromatography MS method.

3) Some typographical errors

Line 140: In this (these) conditions, we estimated that the separation of our 17 classes of lipids

Line 147: Table 1. Chromatographic and mass spectrometry characterization for internal standard and few molecular specie(s).

Line 156. Nevertheless, we observed that sodium adduct [M+Na]+, and fragment source ion  [(M+H)-H2O]+ were majority molecular ion(s) for CE and ceramides respectively.

Line: 281. Sample(s) were extracted immediately after the blood deposit in a tube or on the blotting paper.

Figure 3. Legend provides no text for figure N.

Supplementary Table S4. Supplementary Table 4: Relative quantities (in %) of each molecular species within each class for whole blood, DBS_T3 and DBS_T3 analysis. Means of n=3 meaurement, +/- SEM.

I think DBS3_T3 should be DBS3_T0 and ‘s’ is missing in measurement

Author Response

Thanks to the reivewers comments. Please see attachment.

Reviewer 2 Report

The current article by Le Faouder et al. discusses the utility of SFC-HRMSfor lipidomics analysis using dried blood spot. The study is well conducted and the data presented support the conclusion. However, I think the manuscript can be improved if the following concerns can be addressed - 

  1. the authors addressed the storage time issue of the dried blood spot samples. However, they could not explain some chemical changes in the stored samples. The authors correctly mentioned that a wide range of storage time and temperature effect should be assessed. However, it looks like they only assessed three weeks of storage at room tepmperature. I strongly suggest they test the method on some other storage times and temperatures.
  2. The manuscript should be thoroughly checked for grammar and spelling mistakes. Some examples are below - 

Line 46: occurr

line 63: High resolution mass "spectrometric" approaches

Line 65: unbiases approaches "are"

Line 85: Once the "technique" is chosen

Line 140: "Under these conditions"

There are several other mistakes present throughout the manuscript. 

Author Response

Thanks to the reviewers' comments, please see attachment.

Reviewer 3 Report

The article by Pauline Le Faouder et al. is describing a potentially new approach to how the analyse dry blood spots by SFC-HRMS. Unfortunately, the article is very difficult to read because of the poor English written language. There are many mistakes, and some of the sentences seem doesn’t make sense. I would highly recommend thoroughly review and rewrite the whole manuscript. My other recommendation would be that authors should submit the manuscript to some skilled analytical chemist in the area of chromatographic separations and mass spectrometry. There are many mistakes in terminology, vocabulary and reporting standards. As a novel assay publication, I would expect to see figures showing chromatograms (base peak or total ion current).

Databases of metabolites are usually built using commercially available standards and their analyses by chromatographic assays followed by tandem mass spectrometry to obtain fragmentation spectra. Authors haven’t done anything of that, thus leaving their in-house database just an ordinary list of molecules with no structural confirmation what so ever (or at least it wasn’t mentioned). I would recommend checking the Metabolomics Standard Initiative for guidelines for reporting metabolite identifications. My apologies, but the presented manuscript contains too many unknown variables, and I have to recommend rejecting the manuscript for publication in the current state.

Please find a list of some additional notes and comments:

  • Blood Untargeted Lipidomic profiling on simple dry blood spot by SFC-HRMS – I would suggest “Untargeted lipidomic profiling of dry blood spot by SFC-HRMS” but it is just a suggestion; feel free to ignore it.
  • Please doublecheck the spelling of “technics” throughout the whole manuscript; it is “techniques.”
  • Please use the correct analytical terminology – not - polar & hydrophobic but hydrophilic & hydrophobic. Precise m/z – exact mass (m/z always in Italic)
  • Line 64 – change “not targeted” to “non-targeted.”
  • Line 62 – 64: The sentence sounds weird; please rewrite it. Thank you.
  • Line 74: For the stationary phase of C18 and C8, use lower case for the number of carbons
  • Line 75 – 79: This is not entirely true – many assays are high throughput (15 min); polarity switching is not an issue for data processing. Please do a more thorough literature search.
  • Line 127: Please provide the full names of shortcuts for LC columns.
  • Line 131: What mean to have a “best signal”? Please define your criteria.
  • Line 140: How did you define that separation was “optimal”? Please be more specific.
  • Line 156: “Fragment ion source” is not a standard nomenclature – please use “in-source fragment” instead
  • Line 159: (a little note) Amount of Na+ adducts usually reflects how clean your instrument is. It is not easy to replace it with other ions like NH4+ etc
  • Figure 1: The figure is hard to read; please make it bigger that readers can distinguish peak shapes.
  • Table 1: Please make the table more readable – some of the exact masses have the fourth decimal digit on another line. Complex MS adducts are written only with one bracket – [M+H-H2O]+ also, fix H20 to H2O
  • (…)

Author Response

(The authors gave the same response as above.)

Round 2

Reviewer 2 Report

The authors adequately addressed my concern in their revised version. I would still suggest to include the justification for not including other storage conditions/time in the main text with bit more clarity. Otherwise, the manuscript may be accepted in the present form.

Author Response

Reviewer 2 :

The authors adequately addressed my concern in their revised version. I would still suggest to include the justification for not including other storage conditions/time in the main text with bit more clarity. Otherwise, the manuscript may be accepted in the present form.

We agree totally to the reviewer, we added the justification:

“After discussion with clinicians, we decided to choose the worst situation with the storage of DBS 3 weeks at room temperature indeed these sample are not usually stored in fridge”

to the section 2.4.2

Reviewer 3 Report

Great work on the manuscript it looks much better!!! Thank you for addressing most of the comments and notes. Just a few minor points I found out:

  • in 2.3. Analysis of NIST reference plasma - there is a typo before reference 14 - "Floch" I guess it should be "Folch" LLE
  • This is just a suggestion - Figures 1&2 have a poor quality probably as a result of an image import. To improve the quality of the manuscript you should try to improve the graphic quality of those figures.

Author Response

Reviewer 3 :

Great work on the manuscript it looks much better!!! Thank you for addressing most of the comments and notes.

We thank a lot the reviewer for these nice remarks.

Just a few minor points I found out:

  • in 2.3. Analysis of NIST reference plasma - there is a typo before reference 14 - "Floch" I guess it should be "Folch" LLE

It has been changed in the manuscript

  • This is just a suggestion - Figures 1&2 have a poor quality probably as a result of an image import. To improve the quality of the manuscript you should try to improve the graphic quality of those figures.

We tried to improve this point.